

# Excessive aggregation of fine particles may play a crucial role in adolescent spontaneous pneumothorax pathogenesis

Sibo Wang[1], Jun Li[1], Mengjiao Qian[1], Jing Wang[1], Yongxing Tan[2], Haibo Ou[1], Zhongyin Wang[1], Xiao Chen[1], Yunjiao Tu[2] and Kai Xu[3]

[1] Department of Cardiothoracic Surgery, The Southern Yunnan Central Hospital of Yunnan/The First People's Hospital of Honghe Prefecture, Gejiu, Yunnan, China

[2] Department of Pathology, The Southern Yunnan Central Hospital of Yunnan/The First People's Hospital of Honghe Prefecture, Gejiu, Yunnan, China

[3] Department of Clinical Laboratory, The Southern Yunnan Central Hospital of Yunnan/The First People's Hospital of Honghe Prefecture, Gejiu, Yunnan, China

Corresponding authors
Sibo Wang, wangsibo86@126.com
Jun Li, 13988062023@139.com

## ABSTRACT

**Background**. The pathogenesis of primary spontaneous pneumothorax (PSP) is unclear. Fine particles aggregated in the lung can be phagocytosed by alveolar macrophages (AMs) to induce an inflammatory reaction and damage local pulmonary tissue, which could be a mechanism of PSP. This project aimed to explore the pathological association between fine particulate matter and PSP.

**Methods**. Thirty pulmonary bullae tissues were obtained from surgery of PSP patients (B group). The adjacent normal tissues of the lungs were defined as the control S group. Another 30 normal lung tissues with nonpneumothorax disease (NPD) were applied as the control N group. Hematoxylin and eosin (H & E), Wright-Giemsa (W-G), Victoria blue, and immunohistochemical (IHC) staining experiments were performed to measure the levels of fine particulate matter, alveolar macrophages (AMs), pulmonary elastic fibers, monocyte chemoattractant protein-1 (MCP-1), and matrix metalloproteinase-9 (MMP-9) in the lung tissues. The serum levels of MCP-1 and MMP-9 were prospectively analyzed as well.

**Results**. Histopathological examinations revealed obvious deposition of fine particulate matter and inflammatory reactions (proliferation of AMs) in the B group, compared with those in the S group and the N group. These alterations were significantly associated with PSP. The numbers of AMs and pulmonary elastic fibers, the positive area of the H-score, as well as the concentrations of MCP-1 and MMP-9 in the lungs of the experimental group were obviously raised compared with the controls ($P < 0.05$).

**Conclusions**. Fine particulate matter aggregation, inflammation (macrophage hyperplasia), and overexpression of MCP-1 and MMP-9 may contribute to the pathogenesis of PSP. The overaccumulation of fine particulate matter may play a crucial part in the occurrence of adolescent and young adult PSP.

**Trial registration**. This project was enrolled on the Chinese Clinical Trial Registry: ChiCTR2100051460.

## INTRODUCTION

A primary spontaneous pneumothorax (PSP) is a commonly encountered health problem that typically occurs during leisure time, primarily in tall and thin boys with a low body mass index (*Chiu et al., 2018*; *Sahn & Heffner, 2000*). The incidence of PSP annually is approximately 18–28/100,000 for males and 1.2–6/100,000 for females (*MacDuff, Arnold & Harvey, 2010*). While several known risk factors are associated with PSP, the detail pathogenesis of adolescent spontaneous pneumothorax is still uncovered. Therefore, a deeper understanding of the underlying mechanisms would greatly benefit PSP therapy in clinical settings.

PSP is believed to arise from chronic damage of the subpleural alveolar structure caused by oxidative stress, hypoxia, and chronic inflammation (*Goven et al., 2010*). In addition, pulmonary alveoli formation can cause rupture of alveolar tissue, thereby increasing the permeability of alveolar tissue. A current report has indicated that matrix metalloproteinases (MMPs) play a role in facilitating the formation of bullae and areas of pleural porosity, leading to spontaneous pneumothorax (*Chiu et al., 2018*). Additionally, the infiltration of proinflammatory factors produced by type II pneumocytes and the degradation of elastic fibers have been associated with PSP (*McCourtie et al., 2008*). Moreover, macrophage hyperplasia has been verified as a crucial factor in the chronic inflammatory response of PSP (*Shapouri-Moghaddam et al., 2018*). Epidemiological studies also have demonstrated a close association between the pulmonary deposition of fine particulate matter and the occurrence and progression of various respiratory diseases (*Chang et al., 2015*). In this study, our objective was to investigate the potential involvement of pulmonary aggregation of fine particulate matter in the occurrence of PSP. We obtained pulmonary samples from PSP surgeries and conducted analyses to detect fine particulate matter, examine the histological structure, and assess other inflammatory damage. Our preliminary results suggest that the fine particulate matter may serve as a pathogenic factor contributing to the occurrence of PSP.

## MATERIALS & METHODS

### Groups and specimens

The bullae (B group) and paravesicular (S group, normal adjacent tissue control) lung tissue specimens were obtained from 30 adolescent and young adult PSP patients around 15–29 years old who underwent a surgery of video-assisted thoracoscopic between November 2020 and October 2021 at Southern Yunnan Central Hospital (Honghe First People's Hospital). PSP was characterized as the episode of spontaneous pneumothorax in the patient without any apparent pulmonary disease, inherited disease, or trauma disorder. The clinical characteristics of the patients, including body mass index, age, sex, height, weight, smoking habits, and pathological symptoms, were recorded and analyzed (Table 1). Another 30 normal lung tissues from patients aged younger than 30 years old with nonpneumothorax diseases were obtained as another control (N group). All tissue specimens were immediately collected after the surgery and stored in a freezer at −80 ° C. This study was authorized by the Medical Ethics Management Committee of Southern

**Table 1  Clinical characteristics of the patients with SPS or non-PSP.**

| Characteristic | Patients with PSP (B and S groups, $N = 30$) | Control patients ($N$ group, $N = 30$) | Statistical analysis |
|---|---|---|---|
| Age (years) | | | |
| Mean $\pm$ SD | 21.1 $\pm$ 3.43 | 23.1 $\pm$ 3.52 | $t = 2.038$ $P = 0.051$ |
| Min–Max | 15–29 | 18–30 | |
| Median | 20 | 22.5 | |
| Sex | | | |
|     Female | 3 (10%) | 13 (43.4%) | |
|     Male | 27 (90%) | 17 (56.6%) | |
| BMI (kg/m$^2$) | | | |
| Mean $\pm$ SD | 18.77 $\pm$ 2.68 | 24.1 $\pm$ 3.9 | $t = 5.816$ $P < 0.05$ |
| Min–Max | 15.7–26.7 | 16.9–33.5 | |
| Median | 18.2 | 23.5 | |
| Smoking | | | |
| Yes | 6 (20%) | 10 (33.3%) | |
| No | 24 (80%) | 20 (66.7%) | |
| Height (cm) | | | |
|     Mean $\pm$ SD | 172.3 $\pm$ 6.68 | 160.7 $\pm$ 7.68 | $t = 5.821$ $P < 0.05$ |
| Min–Max | 150–182 | 149–175 | |
| Median | 174 | 161 | |
| Weight (kg) | | | |
|     Mean $\pm$ SD | 55.6 $\pm$ 7.06 | 62.3 $\pm$ 11.4 | $t = 2.554$ $P = 0.016$ |
| Min–Max | 44–70 | 45–90 | |
| Median | 55 | 61.5 | |
| Pathology | | | |
|     Pulmonary bullae | Bullae (30) | – | |
|         Tumor | – | AAH (3), MIA (2), AIS (2) | |
|         Other | – | Benign lesion (5) | |
| | | | Trauma (18) |

**Notes.**
Data are presented as the mean $\pm$ SD or percentage (%) of the patients.
BMI, body mass index; SD, standard deviation.
PSP, primary spontaneous pneumothorax; AAH, atypical adenomatous hyperplasia; MIA, minimally invasive adenocarcinoma; AIS, adenocarcinoma *in situ*.

Yunnan Central Hospital (HH2020LLSC-2), and all people who participated in this study signed informed consent forms.

## Materials

The 5250040 Varioskan Flash Microplate Reader used in this study was from Thermo Scientific (Waltham, MA, USA). The AK-RO-C2 Ultrapure Water Polishing System was from ELKAY (Chicago, IL, USA). The D3024R high-speed refrigerated centrifuge was from DLAB Scientific (Singapore, Singapore). The Nikon Eclipse E100 Orthostatic

Optical Microscope and the Nikon DS-U3 Imaging System were from Nikon (Tokyo, Japan). The RM2016 Pathological Slicer was from Leica (Shanghai, China). The Image-Pro Plus 6.0 was from Media Cybernetics (Rockville, MD, USA). Absolute ethanol was purchased from Sinopharm Chemical (Beijing, China). The Victoria Blue Dye kit and neutral balsam were purchased from Servicebio (Wuhan, China). Normal rabbit serum was from Boster (Wuhan, China). The E0573 oven, pH 6.0 citric acid antigen-repairing solution, phosphate-buffered saline, bovine serum albumin, H&E stain, DAB horseradish peroxidase chromogenic kit, human MMP-9 enzyme-linked immunosorbent assay (ELISA) kit, human chemokine ligand 2 (CCL2)/chemoattractant protein-1 (MCP-1) ELISA kit, and MCP-1/CCL2 rabbit polyclonal antibody were purchased from Beyotime (Shanghai, China).

## H & E staining

Routine sections of experimental lung tissue (with a thickness of approximately 4 μm) were placed in an oven for 4 h (60 ° C), treated with xylene, dehydrated with gradient alcohol (100%, 95%, and 70%, respectively), and rinsed with tap water. The slides were stained by hematoxylin dye for 6 min, rinsed with water for 1 min, differentiated with 0.5–1% hydrochloric acid alcohol for a moment (until the color changed from blue to red three times), and washed with water for more than 15 min. Next, the sections were immersed in 1% eosin alcohol for 1–2 min, rinsed with water for 1 min, and subjected to gradient alcohol (95%, 95%, and 100%, respectively) dehydration for 1 min each. Treatment with carbonate xylene (1:3) was performed for 10 s for dehydration, and treatment with xylene I for 1 min was carried out so that the sections became transparent. After drying, the sections were sealed with neutral gum. All standard H&E-stained samples were analyzed and recorded under a Nikon Eclipse E100 camera imaging capture system. Each stained sample with obvious histologic alteration was analyzed by the same pathological specialist in pulmonary pathology. Each test result was verified by another pathologcal expert. Pathological alterations, including lung histological features (such as bleb/bulla, fine particles, pathologic hyperplasia, granulation tissue, *etc.*) and cellular features (including macrophages, lymphocytes, giant cells, histiocytes, eosinophils, and neutrophils), were carefully recorded.

## Fine particulate matter and alveolar macrophage (AM) examinations

The Swiss Giemsa dye kit (Pinofi Biotechnology, Wuhan, China) was applied for standard Wright-Giemsa (W-G) staining. The W-G-stained slides were observed under a light microscope. Three visual fields were randomly selected under the microscope and observed at 100×, 200×, and 400× magnifications, respectively. The distribution of fine particulate matter was observed using an image analysis system. The number of AMs was calculated from five randomly selected visual fields under a 400× high-power microscope.

## Elastic fiber staining

The sample to be tested was fixed by 4% paraformaldehyde and stained by Victoria blue elastic fiber dye (Servicebio, Wuhan, China). The stained lung tissue was photographed by an Eclipse Ci-L microscope (Nikon, Tokyo, Japan). The target area of the image
was magnified by 200×. The positive area of pulmonary elastic fibers was analysized by Image-Pro Plus 6.0. Three visual fields were analyzed in each slide. The percentage of the positive area (%) was calculated as follows: positive area/tissue area × 100.

## Immunohistochemical (IHC) staining

After formalin fixation, the pulmonary tissue slice (4-μm-thick) was subjected to anti-human MCP-1 (1:100 dilution) polyclonal antibody or anti-human MMP-9 (1:200 dilution) monoclonal antibody IHC staining. The stained slices were analyzed by a Nikon Eclipse E100 Image System (Nikon, Tokyo, Japan). The obvious pathologically altered area was analyzed by two independent pathologists. The quality of the staining was semi-quantitatively evaluated as following: no staining (the positive cells were <10%); weak staining (the positive cells were approximately 10–50%); and strong staining (the positive cells were >50%). The H-score was calculated as follows: $H = \sum(\pi \times I) = $ (% of weakly stained area × 1) + (% of moderately stained area × 2) + (% of strongly stained area × 3), where $\pi$ represents the percentage of the positive signal pixel area and $I$ indicates the positive grade. The H-score value ranges between 0 and 300. A higher H-score indicates a stronger intensity of positive staining.

## ELISA

Based on the manufacturer's protocals provided with the ELISA kits (Beyotime, Shanghai, China), human MCP-1 and human MMP-9 were detected in the lysate of the pulmonary tissue specimens.

## Statistical analysis

SPSS (version 25.0, IBM-SPSS, Chicago, IL, USA) was applied for statistical examination. Graphs (GraphPad Prism, San Diego, CA, USA) was version 8.02. Group comparisons of the histopathological data for the PSP and the non-PSP groups were examined with the Fisher's exact test or chi-squared test. Differences of normally distributed continuous variable data between the PSP patients and the non-PSP controls were detected by the $t$-test. All statistical examines were two-tailed. $P$-value < 0.05 was defined as being significant.

# RESULTS

## PSP tissue had significant alveolar structural damage accompanied by fine particle aggregation and inflammatory reactions

The H&E staining experiments showed that the pulmonary alveolar structure was damaged and fused into bullae. In addition, a large number of dust particle-absorbed macrophages were observed in the alveolar cavity and alveolar interstitium in the B group (experimental group) (Figs. 1C–1D). The Victoria blue dye staining experiments also demonstrated that compared with the S and N groups (Figs. 1G–1H), the B group exhibited dilated alveoli, a widened septum, enlarged alveoli pores, a broken alveoli septum, a reduced pulmonary capillary bed, thickened intima of pulmonary arterioles, increased elastic fibers, and numerous deposited fine particulate matter in the stroma and alveolar cavity accompanied by a large number of AMs (Figs. 1E–1F). Furthermore, the W-G staining experiments showed that compared with the S and N groups, there was obvious aggregation of fine

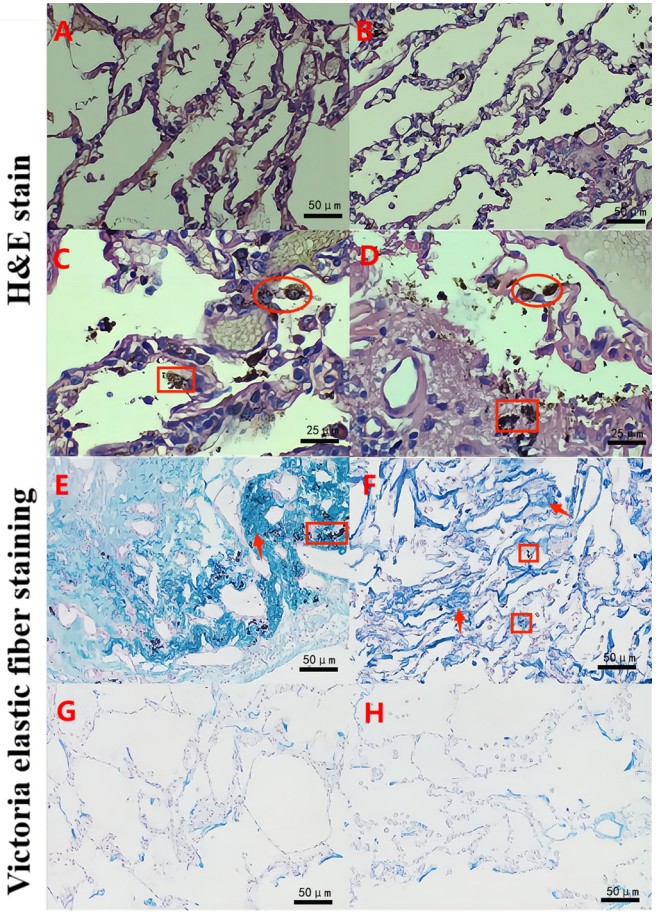

**Figure 1** **The pulmonary structure was damaged in the PSP patients as determined by hematoxylin and eosin (H & E) and Victoria staining.** H&E staining reveals that a clear alveolar structure can be observed in a typical representative image of the lung tissue specimens of the normal adjacent tissue control group (S group; A) and the normal lung tissues from the nonpneumothorax disease group (N group; B). Large numbers of alveolar macrophages (○) and adsorbed dust particles (□) were remarkably observed in the lung tissue specimens of the PSP patients (B group; C–D). Victoria elastic fiber staining was also performed in the B (E–F), S (G), and N (H) groups. Large numbers of particles (□) were deposited in the alveoli, and the number of elastic fibers was significantly increased (↑) in the lung tissue specimens of the PSP patients (E–F). The blue color represents elastic fibers.

particulate matter and increased numbers of AMs in the alveolar wall and alveolar cavity in the B group (Fig. 2). These data suggest that there was fine particle matter aggregation and inflammation, accompanied with a broken pulmonary structure and pulmonary bullae formation in the PSP pulmonary tissue.

## Significantly increased pulmonary elastic fiber content in the PSP patients by Victoria blue dye staining

After counting and calculation of the Victoria blue dye staining results, the pulmonary elastic fiber content in the B group was $10.67 \pm 4.39$ (95% confidence interval (CI) [9.026–12.31]. However, that in the S and N groups was $6.03 \pm 2.45$ (95% CI [5.114–6.943]) and

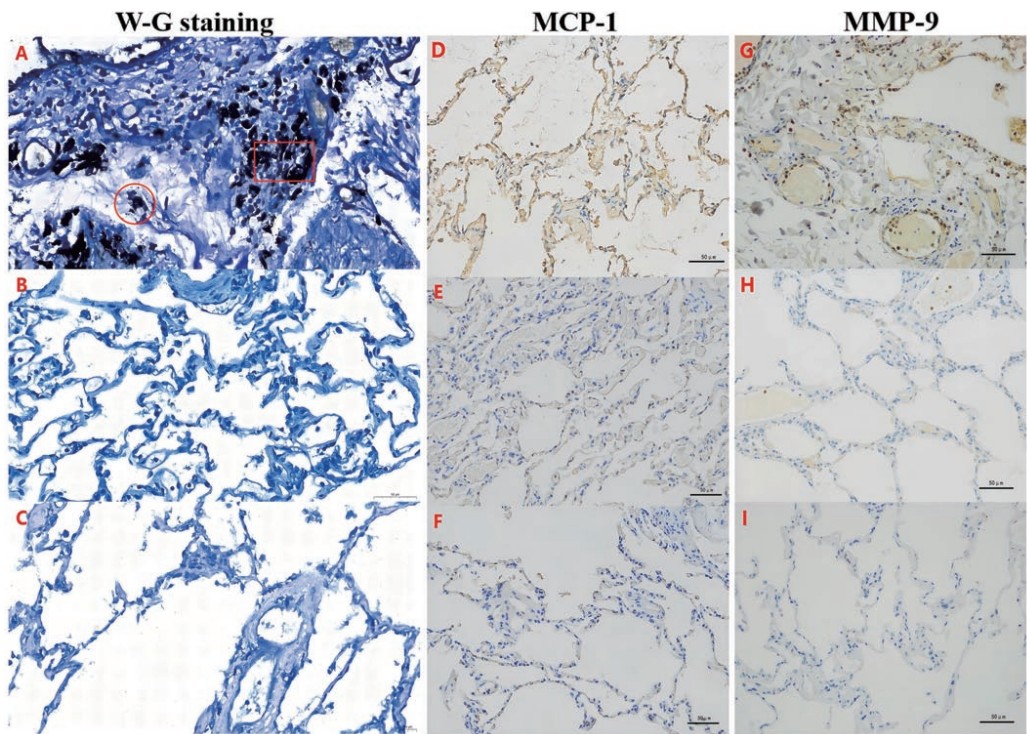

**Figure 2** **MCP-1 and MMP-9 expression was increased in the pulmonary tissues of the PSP patients by Wright-Giemsa (W–G) and immunohistochemical (IHC) staining.** Alveolar macrophages (∘) and fine dust particles (□) were observed by W-G staining (A–C) in the PSP patients (B group; A) and controls (S and N groups; B–C). The expression of MCP-1 (D–F) or MMP-9 (G–I) was examined by IHC staining in the lung tissue specimens of the B group (D, G), S control group (E, H), and N control group (F, I), respectively. Nuclei were colored blue, and the positive expression of DAB was brownish yellow.

5.89 ± 2.92 (95% CI [4.801–6.98]), respectively, indicating that the pulmonary elastic fiber content of the B group was statistically higher than that of the S or N group. The difference between the B group and the S or N group was significant ($P < 0.05$). Nevertheless, no significant difference was found in the pulmonary elastic fiber content between the N and S groups ($P = 0.842$; Fig. 3A).

After staining, five fields of view were randomly obtained from the sections at 400 × magnification. The remarkable histopathological features of the PSP were fine particulate matter (87%), followed by AM accumulation (83%), pathologic hyperplasia (proliferation of fibrous tissue and capillary endothelial cells) (80%), eosinophilic infiltration (77%), hemorrhage (73%), and neutrophil infiltration (67%). The comparisons of the pathological findings between adolescent and young adult PSP (B and S groups) and non-PSP (N group) are presented in Table 2. Compared to the S and N groups, significant elevations of fine particulate matter and inflammation (Fig. 1) were found in the PSP patients ($P < 0.05$). Following, logistic detection also showed that the accumulation of AMs (odds ratio (OR): 0.116; 95% CI [0.034–0.39]; $P < 0.05$) and fine particulate matter (OR: 0.065; 95% CI [0.015–0.211]; $P < 0.05$) was obviously associated with the occurrence of PSP.

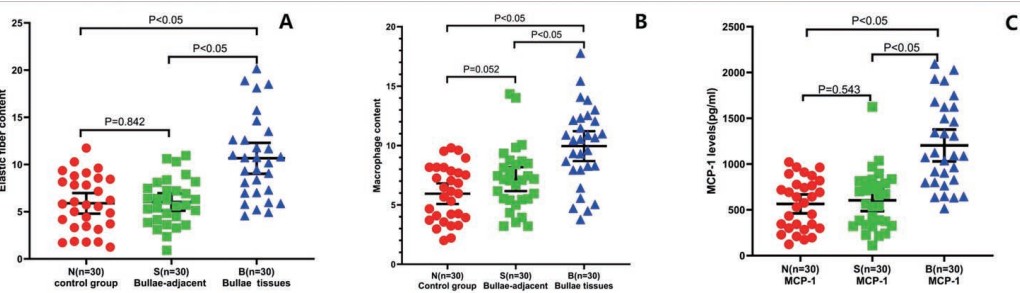

**Figure 3** Alterations of pulmonary elastic fibers, macrophages, and levels of MCP-1 in PSP patients. The pulmonary elastic fiber content (A) and macrophage content (B) were evaluated by H&E staining, Victoria staining, and Wright-Giemsa staining, in the indicated groups. The MCP-1 expression level (C) was examined by ELISA in the PSP (B group) patients, S control group, and N control group. Each data point represents the average of independent assays performed in triplicate.

**Table 2** Pathological characteristics of the patients with adolescent PSP (B and S groups) or non-PSP (N group).

| Pathological characteristics | Patients with PSP (B group) (N = 30) | Patients with PSP (S group) (N = 30) | Control patients (N group) (N = 30) | P-value |
|---|---|---|---|---|
| Tissue features | | | | |
| Fine particulate matter accumulation | 26 (87%) | 9 (30%) | 8 (27%) | <0.05 |
| Bleb/bulla | 30 (100%) | | | |
| Pathologic hyperplasia | 24 (80%) | 17 (57%) | 16 (53%) | <0.05 |
| Hemorrhage | 22 (73%) | 10 (33%) | 12 (40%) | <0.05 |
| Granulation tissue | 10 (33%) | 11 (36%) | 9 (30%) | 0.781 |
| Cellular features | | | | |
| Macrophage accumulation | 25 (83%) | 9 (30%) | 11 (37%) | <0.05 |
| Multinucleated giant cells | 17 (57%) | 7 (23%) | 8 (33%) | <0.05 |
| Eosinophilic infiltration | 23 (77%) | 16 (53%) | 12 (40%) | <0.05 |
| Lymphocyte aggregation | 6 (20%) | 8 (26%) | 8 (26%) | 0.542 |
| Neutrophil infiltration | 20 (67%) | 14 (47%) | 10 (33%) | <0.05 |

**Notes.**
PSP, primary spontaneous pneumothorax.
$P < 0.05$ indicates a significant difference.

## Significantly increased inflammation in the PSP patients

Using the W-G staining method, the number of AMs in each 100 cells was calculated in five randomly selected fields under 400 × magnification. The average number of AMs in the alveolar wall and the alveolar cavity of the B group was $9.95 \pm 3.36$ (95% CI [8.697–11.21]), which was statistically higher than that of the S group ($7.17 \pm 2.69$; 95% CI [6.162–8.171]) and the N group ($5.94 \pm 2.32$; 95% CI [5.072–6.809]; $P < 0.05$) (Fig. 3B). Combined with the findings above (eosinophilic infiltration (77%) and neutrophil infiltration (67%) in PSP patients), these results indicate that there was significant inflammation in the PSP patients.

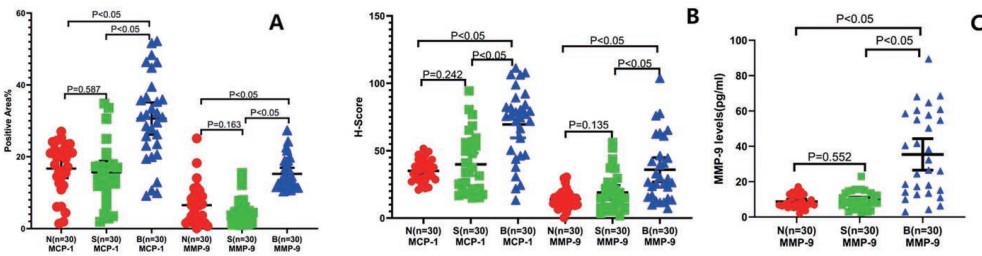

**Figure 4  Pulmonary mesenchymal alterations in PSP patients.** The pulmonary MCP-1- (left three groups) and MMP-9-(right three groups) positive areas (A) and H-score (B) were evaluated by immuno-histochemical staining, respectively, in the indicated groups. The MMP-9 expression levels (C) were examined by ELISA in the PSP (B group) patients, S control group, and N control group. Each data point represents the average of independent assays performed in triplicate.

## Significantly increased MCP-1 and MMP-9 expression levels in the pulmonary tissue of the PSP patients

In order to further analyze the alterations of mesenchyme in lung tissue of the PSP patients, we detected the expression of MCP-1 and MMP-9. IHC examination indicated that the positive area (Fig. 4A; 30.63 ± 12.07%, CI [26.12–35.14]) and the H-score (Fig. 4B; 69.47 ± 26.83, CI [59.45–70.49]) of MCP-1 of the lung tissue in the B group were significantly increased comparing to those in the S group (15.67 ± 8.63%, CI [12.45–18.9]; 39.87 ± 22.25, CI [31.56–48.18], respectively) and the N group (16.72 ± 7.1%, CI [14.07–19.38]; 34.99 ± 8.26, CI [31.9–38.07], respectively) ($P < 0.05$). In addition, the positive area of MMP-9 (Fig. 4A; 15.24 ± 4.39%, CI [13.57–16.91]) and the H-score (Fig. 4B; 35.86 ± 4.26, CI [26.85–44.87]) in the B group were remarkably elevated than those in the S group (4.71 ± 3.60%, CI [3.34–6.08]; 18.98 ± 2.51, CI [13.68–24.28], respectively) and the N group (6.52 ± 5.53%, CI [4.42–8.62]; 14.08 ± 1.21, CI [11.52–16.64], respectively) ($P < 0.05$).

The ELISA results also showed that the MCP-1 (Fig. 3C; 1203.13 ± 84.98 pg/mL, 95% CI [1029–1377]) and MMP-9 (Fig. 4C; 1203.13 ± 84.98 pg/mL, 95% CI [26.5–44.3]) levels in the pulmonary tissue of the B group specimens were significantly increased comparing to those in the S (604.58 ± 57.86 pg/mL, 95% CI [486–723]; 9.43 ± 0.87 pg/mL, 95% CI [7.6–11.2], respectively) and N groups (564.79 ± 50.55 pg/mL, 95% CI [461–668]; 8.69 ± 0.68 pg/mL 95% CI: 7.3–10, separatively) ($P < 0.05$). These data reveal that there are chronic inflammatory reactions in the pulmonary tissue of PSP patients, with significant thickening of mesenchymal tissue, resulting in increased tissue fragility because of an imbalance in the proportion and distribution of extracellular matrix, elastic fibers, and collagen fibers, which could be the histological basis for the easy formation of PSP.

## DISCUSSION

In the last few decades, multiple epidemiological researches have firmly verified that air pollution is tightly associated with pulmonary disease. As the major pollutant, fine particulate matter is closely related to the occurrence of many respiratory diseases (*Goven et al., 2010*). Because the inhaled fine particles are easily deposited in the lung tissue but

are difficult to be eliminated, a series of pathological effects can be induced. Our data provide the evidence. In this study, we performed H&E and W-G staining experiments on the pulmonary tissue from patients with PSP, the adjacent normal pulmonary tissues, and pulmonary tissue from the normal control group. It was found that black particles were deposited in the alveolar cavity and lung stroma in the bulla and the surrounding alveolar wall. On the contrary, it was difficult to find any black particles on the pulmonary surface outside of the lung tip in the control groups. Clinical observations have indicated that most pulmonary bullae that cause a pneumothorax are located in the lung tip. Obvious inflammatory cell exudation, pyoderma formation, and characteristic black spots and flakes are often distributed on the lung surface around the pulmonary bullae (*Lee et al., 2010*).

Due to the relative concentration of industry, air pollution in cities is clearly higher than in rural areas. *Bertolaccini et al.*'s (*2015*) prospective study on the epidemiology of PSP suggests that meteorological parameters and air pollution indicators may be the reasons for the high incidence of PSP in cities. In addition, *Park et al. (2018)* and *Han et al. (2019)* used different methods to analyze the correlations between $O_3$, $NO_2$, particulate matter (PM) 10, and PM2.5 and the occurrence of PSP. They found that these air pollution indicators were obviously associated with the occurrence of PSP. When the concentrations of $O_3$, PM10, $NO_2$, and PM2.5 increased, the incidence rate ratio (IRR) of PSP elevated by about 15, 3, 16, and 5 times, separatively (*Park et al., 2018*). Their conclusions all show a significant correlation between an increase in the atmospheric pollutant concentration and an increase in the IRR of PSP. Although our study did not conduct such research analysis, *Bertolaccini et al. (2015)*; *Park et al. (2018)*; *Han et al. (2019)* have provided indirect evidence for our findings from an epidemiological perspective.

As the primary phagocyte, AMs play a vital role in the local immune defense by initiating the local inflammatory response. Thus, AMs can be used as an indicator to evaluate the pulmonary inflammatory response (*Chiu et al., 2017*). Our observation reveals that the accumulation of a large number of AMs and inflammatory cell infiltration were clearly observed. These alterations can lead to the narrowing and breaking of the alveolar septum, causing the normal structure to be lost and the formation of pulmonary blisters. Indeed, other research results suggest that the chemotaxis, adhesion, phagocytosis, and secretion of cytokines from AMs are the key mediators playing a role in the natural immune function (*Chang et al., 2015*; *Chen et al., 2021*). AMs can migrate and gather in the inflammatory area through chemotaxis, and they are activated after swallowing foreign bodies to secrete a variety of active substances, which further regulate the physiology and pathology of the local lung.

Other research results suggest that the chemotaxis, adhesion, phagocytosis, and secretion of cytokines from AMs are the key mediators playing a role in the natural immune function (*Chang et al., 2015*; *Chen et al., 2021*). AMs can migrate and gather in the inflammatory area through chemotaxis, and they are activated after swallowing foreign bodies to secrete a variety of active substances, which further regulate the physiology and pathology of the local lung. MCP-1 is a major component of cysteine-cysteine chemokines, which are secreted by various immune and nonimmune cells, including alveolar macrophages. Most

of the MCP-1 protein is secreted by AMs in the lung tissue and performs a crucial role in the infiltration and activation of AMs. Therefore, AMs can be applied as an important index to detect subacute and chronic inflammatory damage (*Chen & Wang, 2021*; *Di Stefano et al., 2018*). The termination of lung development has been studied extensively, however, no definitive conclusion has been obtained. Some scholars have suggested that from two years old to young adulthood, the pulmonary compartment develops at the ratio of the lung volume, while others have reported that it grows continuously until 8–11 years old (*Burri, 2006*; *Emery & Wilcock, 1966*). In addition, an increase of MCP-1 expression promotes the accumulation of AMs in the airway and respiratory burst, thus strengthening the local inflammatory response and oxidative damage as well as finally leading to airflow obstruction and destruction of the lung tissue structure, especially during lung development in adolescents due to the characteristic of adolescent lung tissue development that the number of alveoli is relatively increased. IHC staining and ELISA experiments in this study showed that the expression of MCP-1 in the PSP group was remarkably greater than that in the S or the N control group. The abnormal aggregation of fine particulate matter may stimulate and induce a high MCP-1 expression and assembly of AMs at the local lung tissue to cause chronic inflammatory reactions and oxidative damage. These pathological alterations would gradually destroy the normal alveolar structure, resulting in thinning and breaking of the local alveolar wall and, finally, leading to the reconstruction of normal alveolar tissue and the formation of pulmonary bullae. After local inflammation, the bullae rupture and produce a pneumothorax. In fact, in our histological observation, pulmonary mesenchyme thickening, local inflammatory reactions, alveolar rupture, and bulla formation were clearly seen in the PSP tissue.

Chronic destruction of subpleural alveolar structures is believed to be the pathological reason for PSP, which can be caused by hypoxia, chronic inflammation, and oxidative stress. Although airway inflammation, abnormal connective tissue, ischemia, and hypoxia are possible reasons, the pathogenesis of PSP still remains uncovered (*Tschopp et al., 2015*). PSP is reported to be caused by the destruction of visceral pleural mesothelial cells due to the increase of the inflammatory elastic fibrosis layer accompanied by increased porosity, which can be easily ruptured (*Noppen et al., 2006*). Victoria staining also revealed that abnormal wavy and worn elastic fibers were located around the pulmonary bullae in the experimental group, pigment-filled macrophages were gathered in the alveolar space, and the abnormally aggregated elastic fibers in the pulmonary bullae had a mound-like contour. In the control groups, the alveolar wall was fibrous with a clear outline. However, the detailed processes of alveolar formation, as well as the association between pleura and the alveolar cavity, are still not clear (*Noppen, 2010*).

Elastolysis in pulmonary disorders happens because of an imbalance between proteases and anti-proteases. MMPs belong to the zinc-dependent enzyme family and are classified into multiple subgroups: stromelysins, collagenases, gelatinases, elastases, and membrane-type MMPs (*Craig et al., 2015*). Although MMPs participate in many physiological processes, including tissue remodeling, wound healing, and angiogenesis, the overexpression of MMPs is associated with some pulmonary disorders (*Oikonomidi et al., 2009*). In our analysis, a significant elevation of elastic fibers was found in SPS lung

tissue. IHC and ELISA experiments further suggested that the protein level of MMP-9 was obviously up-regulated compared with that of the control group. Furthermore, there was a strong correlation between the increase in MMP-9 and the appearance of PSP. PSP is raised by the rupture of the visceral pleura or pulmonary parenchyma. It is associated with weakening of the subpleural pulmonary tissue. Currently, MMP-9 and MMP-2 have been reported to be essential factors for the clinical pathology of pulmonary disorders (*Wang et al., 2015*). MMP-9 belongs to the gelatinase family, which degrades elastic matrix and type IV and V collagen (*Chiu et al., 2018*). Once the inflammation has lost control, alveolar monocyte macrophages are gradually activated to release MCP-1. As a special cytokine, MCP-1 can increase the number of cells that secrete MMP-9, which can selectively destroy the relevant components of the extracellular matrix needed to remodel the pulmonary structure.

However, there are some limitations in this study. Due to ethical issues, it was impossible to obtain healthy normal lung tissue from adolescent healthy volunteers as a normal negative control. We took the non-neoplastic tissue of patients with stage I lung adenocarcinoma and the lung tissue of patients with trauma as a negative control, which could have introduced some bias. Since we only obtained tissue samples from an injured pulmonary site, it was difficult to determine whether the findings in this study are local or systemic. In addition, there were only a limited number of PSP patients who required immediate surgical treatment due to a pneumothorax. Thus, these results may not be applicable to all patients with PSP, especially those who do not accept surgery. More detailed and extensive research is needed.

## CONCLUSIONS

Although the pathogenesis of adolescent and young adult PSP is not clear (*Chiu et al., 2014*; *Lee et al., 2010*), there may be a variety of congenital and acquired complex factors. The results of this study show that the lung tip with a concentrated amount of fine particulate matter is also the most common site of pulmonary bullae. Therefore, fine particulate matter may be associated with the pathogenesis of adolescent and young adult PSP. The mechanism could be that during the development of adolescents, harmful fine particulate matter is inhaled and deposited in the lung tip tissue, which stimulates AMs and induces the high expression of MCP-1 and MMP-9 in the local lung tissue, causing a chronic inflammatory reaction and oxidative damage. After local infection, the bullae rupture and produce a pneumothorax. Thus, controlling environmental pollution, avoiding the inhalation of harmful particles, and actively preventing respiratory tract infection may effectively block the recurrence of PSP in adolescents.

## ACKNOWLEDGEMENTS

The authors wish to thank the critical review of the manuscript and helpful discussion by Dr. Xuening Hu (Department of Thoracic Surgery, Zhongnan Hospital of Wuhan University, Wuhan, China).

### Funding

This work was supported by the Qingdao University Medical Group, Qingdao University, China (Grant numbers YLJT20202005). The funders had no role in study design, data collection and analysis, decision to publish, or preparation of the manuscript.

### Grant Disclosures

The following grant information was disclosed by the authors:
Qingdao University Medical Group, Qingdao University, China: YLJT20202005.

### Competing Interests

The authors declare there are no competing interests.

### Author Contributions

- Sibo Wang conceived and designed the experiments, performed the experiments, analyzed the data, prepared figures and/or tables, authored or reviewed drafts of the article, and approved the final draft.
- Jun Li conceived and designed the experiments, analyzed the data, authored or reviewed drafts of the article, and approved the final draft.
- Mengjiao Qian conceived and designed the experiments, performed the experiments, analyzed the data, authored or reviewed drafts of the article, and approved the final draft.
- Jing Wang conceived and designed the experiments, authored or reviewed drafts of the article, and approved the final draft.
- Yongxing Tan analyzed the data, authored or reviewed drafts of the article, and approved the final draft.
- Haibo Ou analyzed the data, prepared figures and/or tables, and approved the final draft.
- Zhongyin Wang analyzed the data, prepared figures and/or tables, and approved the final draft.
- Xiao Chen performed the experiments, prepared figures and/or tables, and approved the final draft.
- Yunjiao Tu performed the experiments, prepared figures and/or tables, and approved the final draft.
- Kai Xu performed the experiments, prepared figures and/or tables, and approved the final draft.

### Human Ethics

This study was authorized by the Medical Ethics Management Committee of Southern Yunnan Central Hospital (HH2020LLSC-2), and all people who participated in this study signed informed consent forms.

### Data Availability

The raw measurements are available in the Supplementary File.

## Supplemental Information

Supplemental information for this article can be found online at http://dx.doi.org/10.7717/peerj.16484#supplemental-information.

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
