# Peer review of "Excessive aggregation of fine particles may play a crucial role in adolescent spontaneous pneumothorax pathogenesis"

_PeerJ, doi:10.7717/peerj.16484_

## Round 0.1 · original submission · Major Revisions

Please follow the requested revisions in detail.

Reviewer 1 ·

Basic reporting

I reviewed several papers on inflammation and microplastic aggregation and honestly, this one is the best. So sorry authors didn't realize it was on the inflammation and microplastics.

Experimental design

Your experimental design is confusing; control groups should be organised differently. Staining techniques are not clear, specifically - it is not clear why were used three basic staining. The role of inflammation was not elaborateed enough, neither was the jature of particles specific enough. Inflammation is more like accidentally mentioned as in lns. 81. and 180. instead making it linchpin. Also, note the difference between "aggregation" and "accumulation". Particles, as solids, increase in number. Therfore - aggregate.

Validity of the findings

N/A

Additional comments

N/A

·

Basic reporting

Authors have presented the study clearly. Quality of English is good and professional. Also, authors have provided the article structure, figures, tables. Raw data. Additionally, enough context is provided.
However, some small details can be added.

Experimental design

The research question in the study is well defined, relevant and important to the field. It provides important results related to etiology and underlying pathological mechanisms of PSP. Investigation methods are robust. Lastly methods provide sufficient details to replicate the study.

Validity of the findings

Authors have provided the data which appears to be statistically sound. Conclusion support the result and answer the basic research question introduced in the introduction section of the study.

·

Basic reporting

Line 63: Please include a sentence on how formation of bullae increases pleural porosity.
Line 79: "... 30 adolescent PSP patients aged 15-29 years...". Please modify to read '... adolescent and young adult patients...'. People in their 20s are not adolescents anymore.
Line 120: "... the same specialist in pulmonary pathology...". Since microscopic examination includes a variable component of subjectivity, it would have strengthened the conclusions of the paper if more than one pathologist had interpreted the microscopy. The conclusions of your paper are still valid, but 2 reviewing pathologists would have been better.
Line 195: "... pathological hyperplasia (80%)...". Pathological hyperplasia of what? Of fibrous tissue; interstitial connective tissue; type II pneumocytes; pulmonary macrophages; other? Please specify.
Line 231: "... resulting in increased tissue fragility...". Please include a sentence on how thickening of the mesenchymal tissue results in uncreased tissue fragility.
Line 246: "Obvious inflammatory cellulose exudation...". 'Cellulose' is the incorrect word here. Do you mean 'cellular' or 'cytokine' or something else? Please correct.
Line 266: "... including vesicular macrophages...". Do you mean 'alveolar' macrophages?
Line 270: "... and respiratory explosion...". Not clear what you mean by "respiratory explosion"? Maybe 'oxidative stress' or 'respiratory chain over-expression' or something else? Please correct.
Line 272: "... especially lung development...". Do you mean '... especially during lung development'?
Line 273: Please include a sentence on how the lung still develops by the time a person is an adolescent.
Line 280: "After local infection...". Do you mean 'After local inflammation'? There is no presumed 'infection' in PSP, as described in your paper.
Line 282: "... rupture, and alveolar formation...". Do you mean 'bulla formation'? If not, please explain "alveolar formation".
Line 292: "... a mound-like contour diffuse reaction." The words "diffuse reaction" don't seem to belong. Your sentence would be complete by dropping the words "diffuse reaction": '... the abnormally accumulated ... had a mound-like contour'.
Line 314-315: "... it was impossible to obtain healthy normal lung tissue from the PSP patients...:. Why not? Would there not have been more normal-looking lung tissue in areas outside of the bullae, in the uninvolved pulmonary parenchyma? Of course, if the surgical line of resection or staples was very close to the base of the bulla, there would not be much if any normal lung tissue. Please specify.
Line 321: "... due to a pneumothorax attack." Please use a word other than "attack", since a ptx is not an 'attack'. Maybe 'episode' might be more suitable.

Experimental design

Very elegant study. Thorough analysis of the topic. Nice rundown of the currently known pathogenesis of PSP. Good research question and answer. Extensive workup of the pathological tissues (bullae): microscopy, special stains, immunohistochemistry, assessment of alveolar macrophages and MCP-1 and MMP-9. High technical standard. Convincing graphs. Excellent microscopic pictures. Good tables. Extensive and relevant references.

One question the authors may want to elaborate on further in the paper, if data are known and available: Are there more cases of PSP (per capita) in populations residing in cities with high(er) levels of air pollution, as compared to those living in rural settings? Such findings would support the contributing role of particulate air pollutants to the formation of bullae and PSP.

Validity of the findings

Thorough assessment of all new data, statistically validated and with good controls. Scientifically valid conclusions. Limitations of the study are stated and addressed, encouraging further research in this topic.

Additional comments

Welcome addition to the theories on PSP bulla formation. Definitely contributes to the understanding of the pathogenesis of PSP. The detailed workup of the materials used and the topic itself is impressive and likely required a lot of time and effort. The authors are to be congratulated on their undertaking.

---

## Round 0.2 · accepted · Accept

The authors properly addressed the requests and suggestions of the reviewers. Congratulations!

Reviewer 1 ·

Basic reporting

This version of manuscript is much improved, I have mo additional issues.

Experimental design

-

Validity of the findings

-

Additional comments

-

·

Basic reporting

Quality of writing is satisfactory. Authors have provided sufficient amount of context. quality of figures, tables is good as well.

Experimental design

Research lies within the aims and scopes of the journal. Research question is relevant.

Validity of the findings

Results of the findings are replicable. Overall this study is robust and sound.

Additional comments

From my perspective, authors have made the changes as I suggested. In my opinion, this paper is suitable for publication.

·

Basic reporting

I have previously submitted a review. The corrected version is OK as is.

Experimental design

The corrected version is OK as is.

Validity of the findings

The corrected version is OK as is.

Additional comments

The corrected version is OK for publication.